# Effects of Pre-Donated Autologous Blood Transfusion on Peri-Operative Hemoglobin Concentration and Mid-Term Health Outcomes in Primary Total Knee Arthroplasty

**DOI:** 10.3390/jcm11082252

**Published:** 2022-04-18

**Authors:** Jun Tomura, Daichi Morikawa, Masahiko Nozawa, Muneaki Ishijima, Sung-Gon Kim

**Affiliations:** 1Department of Orthopaedic Surgery, Juntendo University Nerima Hospital, 2-1-1 Hongo, Bunkyo-ku, Tokyo 113-0033, Japan; j.tomura.te@juntendo.ac.jp (J.T.); nozawa@juntendo-nerima.jp (M.N.); skim17@me.com (S.-G.K.); 2Department of Orthopaedics, Faculty of Medicine, Juntendo University, 2-1-1 Hongo, Bunkyo-ku, Tokyo 113-0033, Japan; ishijima@juntendo.ac.jp

**Keywords:** autologous blood transfusion, total knee arthroplasty, blood loss, mortality, cardiovascular event

## Abstract

The effects of auto-BT in primary TKA on the perioperative hemoglobin (Hb) concentration and mid-term health outcomes are unknown. This study was performed to analyze the detailed changes in the perioperative Hb concentration before and after the operation (days 0–14 postoperatively), cardiovascular events, and mortality rate within 1 and 5 years postoperatively. One hundred patients undergoing primary TKA with auto-BT using 800 mL of preoperatively collected blood at the authors’ institution were included. The mean Hb concentration before and after autologous blood collection was 12.7 ± 1.1 and 11.7 ± 1.2 g/dL, respectively. After primary TKA with auto-BT, the mean Hb concentration on day 0, 1, 3, 7, and 14 was 10.2 ± 1.2, 9.9 ± 1.2, 10.4 ± 1.3, 10.5 ± 1.3, and 11.0 ± 1.3 g/dL, respectively. Only one (1%) patient required additional allogenic blood transfusion. No patients developed cardiovascular events, and the 1- and 5-year postoperative mortality rate was 1.0% and 2.0%, respectively. Primary TKA with auto-BT showed relatively small perioperative changes in the Hb concentration, a low incidence of cardiovascular events, and a low mortality rate within 1 and 5 years postoperatively. These findings suggest that auto-BT, in which blood is preoperatively collected, is beneficial for patient safety and health, even if its cost-effectiveness may be debatable.

## 1. Introduction

Total knee arthroplasty (TKA) is a well-established and frequently performed procedure for patients with primary or secondary osteoarthritis (OA) or rheumatoid arthritis. The efficacy and safety of TKA have been widely recognized [1]. However, several issues may be encountered during the perioperative period of TKA, such as blood loss, acute or chronic infection, deep vein thrombosis, or bone fracture [2].

Blood loss is a serious problem related to TKA. Although surgeons have attempted to reduce blood loss using several techniques, such as minimally invasive surgery and intra-articular (IA) or intravenous (IV) injection of tranexamic acid (TXA) [3], blood loss frequently results in the need for blood transfusion. Three types of blood transfusion are performed during TKA: allogenic blood transfusion (allo-BT), autologous blood transfusion (auto-BT), and blood salvage (BS).

Allo-BT is generally performed on the day of surgery or the day after surgery if the hemoglobin (Hb) concentration falls below 7.0 to 9.0 g/dL or if symptoms of anemia appear [4,5,6]. Although allo-BT is useful for treating blood loss during surgery, it has several problems, such as issues with blood type incompatibility, limited resources, a risk of blood-borne infection, transfusion-reaction, and alloimmunization [7,8]. Furthermore, these problems can be exacerbated in elderly patients [5,7,8]. Especially in recent years, the number of blood donors has decreased, and the blood stock has been almost eliminated in some countries because of the worldwide spread of coronavirus disease 2019 (COVID-19) [9,10]. Diagnosis of COVID-19 in a blood donor after donation has also been reported, and sufficient blood supply management has not been established [9,10,11].

BS is a transfusion method involving the use of blood that is intraoperatively or postoperatively collected from the patient by draining a joint [12]. The drained blood is reinfused through filters as soon as possible after collection to prevent coagulation and infection. Like allo-BT, BS has several problems, such as a risk of blood-borne infection due to bacterial contamination, high expense, volume limitations, and time consumption [13,14]. Furthermore, BS can reportedly increase the postoperative blood drainage, caused by reinfusion of unwashed salvaged blood containing tissue plasminogen activator [15,16].

Auto-BT, in which blood is preoperatively collected, has advantages over both allo-BT and BS with respect to blood type incompatibility, the risk of transfusion-transmitted infection such as hepatitis virus, volume limitations, and adverse reactions [7,17,18]. However, auto-BT requires the collection of as much blood before surgery as is expected to be lost during surgery, and the rate of transfusion has decreased in recent years [18,19]. The effects of auto-BT in primary TKA on the perioperative Hb concentration and mid-term health outcomes are unknown, although some studies have shown a relationship between TKA and mortality or cardiovascular events [1,20,21].

This study was performed to analyze the effects of blood collection and auto-BT on changes in the Hb concentration during the preoperative and postoperative periods, cardiovascular events, and the mortality rate within 1 and 5 years after TKA using auto-BT.

## 2. Materials and Methods

### 2.1. Ethical Approval

This study was approved by the ethics committee of Juntendo University Nerima Hospital (No. 27–38).

### 2.2. Patient Selection and Data Collection

This was a retrospective study. From January 2012 to December 2014, senior surgeons performed primary unilateral TKA in 100 patients at the authors’ institution. We included the patients who underwent TKA with auto-BT. All of the 100 patients donated 800 mL of blood, and were included in this study. No patients were excluded. The patients comprised 11 men and 89 women, with a mean age of 76.4 ± 6.2 years (range, 55–81 years).

### 2.3. Surgical Methods and Postoperative Treatment

All patients underwent TKA with general anesthesia and a femoral nerve block. We performed unilateral cemented TKA (posterior-stabilized, *n* = 58; cruciate-retaining, *n* = 37; cruciate-substituting, *n* = 5) using a medial parapatellar or midvastus approach with a femoral tourniquet (300 mmHg). The types of implants used were the P.F.C. SIGMA (Depuy Synthes, Raynham, MA, USA) in 90 patients, Bi-Surface (Kyocera Medical Corporation, Osaka, Japan) in 3, Triathlon (Stryker Corporation, Kalamazoo, MI, USA) in 3, Vanguard (Zimmer Biomet, Warsaw, IN, USA) in 2, and Columbus (B/Braun Aesculap Implant Systems, Center Valley, PA, USA) in 2. A suction drain was placed in the joint. For postoperative pain management, the patients underwent placement of femoral nerve catheters with continuous infusion of 0.2% ropivacaine and oral administration of nonsteroidal anti-inflammatory drugs or acetaminophen. The femoral nerve catheter and suction drain were removed 2 days after surgery. All patients received antithrombotic treatment (edoxaban, a factor Xa inhibitor) for 2 weeks from the day after the operation.

### 2.4. Auto-BT

Autologous blood (400 mL of blood) was collected twice (total of 800 mL) from patients whose Hb concentration was ≥10 g/dL during the 35 days before the operation. If a patient’s serum Hb concentration was <10 g/dL, we did not collect any blood, and used allo-BT for blood loss requiring blood transfusion. After the first blood collection, we administered an injection of recombinant human erythropoietin (24,000 units), and gave the patient a daily oral iron supplement (100 mg/day of elemental iron). We allowed at least 1 week between the first and second blood collections, and performed the second collection at least 1 week before the operation. We returned 400 mL of autologous blood at the end of the operation and the day after surgery. If the patient’s Hb concentration was <7 g/dL or they showed serious signs of anemia, we added allo-BT after auto-BT.

### 2.5. Perioperative Blood Tests

Blood tests were performed before the first and second blood collections outpatiently, and at 1 or 2 days before the operation (day of admission), and postoperative days 0, 1, 3, 7, and 14 during hospitalization. The reduction rate of the Hb concentration after the operation was analyzed. The total blood loss was calculated as follows:

Total blood loss (mL) = blood volume × (Hbpre − Hbpost, day1, day3)/Hbpre + transfused blood (mL);

Blood volume (mL) = body weight (kg) × 70 mL/kg.

### 2.6. Mortality Rate and Cardiovascular Events after TKA

To assess the safety of our procedure, we investigated the cardiovascular events and mortality rate within 1 and 5 years after surgery for all patients based on their medical records when the patients returned for the follow-up to our hospital. If the patient did not return to our hospital, we contacted them via a telephone survey.

### 2.7. Statistical Analysis

A Friedman test was run to determine if there were differences in postoperative Hb concentrations. Pairwise comparisons were performed with a Bonferroni correction for multiple comparisons. A P value of <0.05 was considered statistically significant. All statistical analyses were performed with EZR (Saitama Medical Center, Jichi Medical University, Saitama, Japan), which is a graphical user interface for R (The R Foundation for Statistical Computing, Vienna, Austria). More precisely, it is a modified version of R commander designed to add statistical functions frequently used in biostatistics [22].

## 3. Results

Among the 100 patients, 94 underwent TKA for treatment of primary OA, and 6 underwent TKA for treatment of rheumatoid arthritis. The patients’ data are shown in Table 1.

All patients donated and received 800 mL of blood for auto-BT. All patients underwent all blood tests before and during hospitalization, and had the information of cardiovascular event collected within 1 year after TKA and mortality collected within 1 and 5 years after TKA. The mean Hb concentration before the first donation and after the second donation was 12.7 ± 1.1 and 11.7 ± 1.2 g/dL, respectively (Table 2). On day 0, 1, 3, 7, and 14 after primary TKA with auto-BT, the mean Hb concentration was 10.2 ± 1.2, 9.9 ± 1.2, 10.4 ± 1.3, 10.5 ± 1.3, and 11.0 ± 1.3 g/dL, respectively (Table 2). The lowest mean Hb concentration was 9.9 g/dL, and the maximum reduction in the mean Hb concentration compared with that before the operation was 1.8 g/dL on day 1 after surgery. The Hb concentration recovered to 11.0 g/dL on day 14 after surgery. The Hb concentration on postoperative day 3 was significantly increased from that on day 1, and that on day 7 was significantly increased from that on day 0 (Figure 1). The rates of reduction in the perioperative Hb concentration compared with the day before the operation are shown in Figure 2. The lowest rate was 84.6% on day 1 after surgery, and this rate recovered to 93.9% on day 14 after surgery (Figure 2).

Only one (1%) patient in our series required additional allogenic transfusion. In this patient, who underwent surgery for treatment of primary OA, the Hb concentrations on postoperative days 1 and 3 were both 6.0 g/dL, and we added 400 mL of allo-BT in addition to the scheduled auto-BT. The patient continued to show pancytopenia and a fever of unknown origin 6 weeks after the operation, suggesting that the patient might have had a primary illness influencing their perioperative anemia.

With respect to health outcomes, no patients developed cardiovascular events within 1 year after TKA. One patient died of sepsis within 1 year after surgery, and another patient died of cholangiocarcinoma within 5 years. The 1- and 5-year postoperative mortality rate was 1.0% and 2.0%, respectively.

## 4. Discussion

In the present study, we analyzed the changes in the Hb concentration during the perioperative period, and assessed the cardiovascular events and mortality rate after TKA with auto-BT to clarify the usability of auto-BT for TKA. To our knowledge, this is the first study to analyze the changes in the Hb concentration during TKA using only auto-BT (Table 3).

Among days 1 to 14 after primary TKA with auto-BT, the lowest mean Hb concentration was 9.9 g/dL on postoperative day 1 (Table 2 and Figure 1), and only 1 (1%) of 100 patients required additional allo-BT. In TKA without auto-BT, Ha et al. [23] reported that the lowest mean Hb concentration among postoperative day 1, day 2, and the final follow-up was 8.9 g/dL on postoperative day 2, and 36.1% of patients required additional allo-BT (mean volume, 278.9 mL). In another study of TKA without auto-BT, Cip et al. [14] reported that the lowest mean Hb concentration among postoperative days 1, 3, and 5 was 10.2 g/dL on postoperative day 3, and 33% of patients required allo-BT (mean volume, 2.1 units). In TKA with BS, Ha et al. [23] reinfused blood (mean volume, 487 mL) collected from an intra-articular drain using an autotransfusion system; the lowest mean Hb concentration among postoperative day 1, day 2, and the final follow-up was 10.5 g/dL on postoperative day 2, and 11.1% of patients required additional allo-BT (mean volume, 64.4 mL). Cip et al. [14] reinfused collected blood (mean volume, 283 mL) using an autotransfusion system; the lowest mean Hb concentration among postoperative day 1, day 2, and the final follow-up was 9.9 g/dL on postoperative day 3, and 33% of patients required additional allo-BT (mean volume, 2.1 units). Blazekovic et al. [4] also reinfused collected blood (mean volume not reported) using an autotransfusion system; the lowest mean Hb concentration among postoperative days 1, 2, and 5 was 9.2 g/dL on postoperative day 2, and 33% of patients required additional allo-BT. Furthermore, Blazekovic et al. [4] performed BS with an autotransfusion system and auto-BT (mean volume not reported); the lowest mean Hb concentration among postoperative days 1, 2, and 5 was 9.3 g/dL on postoperative day 5, and 16.7% of patients required additional allo-BT (mean volume not reported). In recent years, many clinical studies revealed that the use of TXA reduces blood loss and transfusion rate after TKA [3]. A previous meta-analysis showed that the percentage of patients who required additional allo-BT in TKA with IA and IV injection of TXA were 6.2% and 6.6%, respectively [3]. A randomized controlled trial showed that the percentage of the patients who required additional allo-BT in TKA with IV injection and IV injection of TXA followed by oral TXA were 2% in each group [25]. On the other hand, a case-control study showed that the percentage of the patients who required additional allo-BT in TKA with IV injection and IA injection of TXA were 0.3% and 0%, respectively [26]. In the present study, we showed that the percentage of the patients who required additional allo-BT in TKA with auto-BT was 1.0%. Although it is difficult to compare the rate of additional allo-BT due to several variabilities, such as patient’s background, surgical technique, or postoperative care, TKA with auto-BT may not be inferior to other methods in terms of avoiding allo-BT.

In the present study, the maximum reduction in the Hb concentration from the day before surgery was 1.8 g/dL on postoperative day 1 (Table 2 and Figure 1). Ha et al. [23] reported that the maximum reduction in the Hb concentration compared with before surgery was 4.0 g/dL on postoperative day 2 in TKA without auto-BT. Cip et al. [14] also reported that the maximum reduction in the Hb concentration compared with before surgery was 4.1 g/dL on postoperative day 3 in TKA without auto-BT. Recently, Cho et al. [24] reported the maximum reduction in the Hb concentration compared with before surgery was 4.5 g/dL on postoperative day 3 in TKA without auto-BT. In TKA with BS (average reinfusion volume, 487 mL), Ha et al. [23] reported that the maximum reduction in the Hb concentration compared with before surgery was 2.6 g/dL on postoperative day 2. Cip et al. [14] also reported that the maximum reduction in the Hb concentration compared with before surgery was 3.7 g/dL on postoperative day 3 in TKA with BS (average reinfusion volume, 283 mL). Blazekovic et al. [4] performed BS with or without allo-BT, and reported that the maximum reduction in the Hb concentration compared with before surgery was 5.0 to 5.6 g/dL. In TKA with TXA injection, Cho et al. [24] reported that the maximum reduction in the Hb concentration compared with before surgery was 3.7 g/dL on postoperative day 3 in TKA with IA injection of TXA, and Chang et al. [25] reported that the maximum reduction in the Hb concentration compared with before surgery was 3.5 g/dL on postoperative day 4 in TKA with IV injection of TXA. These results suggest that TKA with auto-BT might prevent a rapid decrease in the Hb concentration during the perioperative period of TKA. The reduction of the Hb concentration in TKA with BS may be attributed to dilutional coagulopathy from large volume, or providing only RBC supplement and washing away the clotting factors and platelets in BS [27]. Avoiding a rapid decrease in the Hb concentration may allow for safer and more effective rehabilitation because of an improved oxygen supply to the muscle and heart.

In addition, important findings are that the lowest Hb concentration was shown on postoperative day 1, and the Hb concentration on postoperative day 3 was significantly increased from day 1 in this study (Table 2 and Figure 1). On the other hand, the previous studies reported the day of the lowest Hb concentration was after postoperative day 3 (Table 3). The rapid recovery from reduced Hb concentration in our study may contribute to a beneficial effect for a low incidence of cardiovascular events, and a low mortality rate after TKA.

Focusing on the use of TXA in TKA, although it has been reported that oral TXA is not recommended, both perioperative IA and IV injection of TXA are widely used, and are safe procedures to reduce blood loss [24,25]. However, TKA with IA or IV injection of TXA showed a higher rate of additional allo-BT and reduction of Hb concentration than those of our procedure [3,24,25]. Further consideration is needed for both patient safety and cost-effectiveness.

In terms of the economic aspect, the total cost of 800 mL of Auto-BT per TKA is JPY 50,000 (approximately USD 450) in our hospital. On the other hand, it has been reported that the cost of IV and IA injection of TXA was USD 78.28 and USD 39.14, respectively [26]. Though auto-BT has a disadvantage in the economical aspect, it may provide advantages to decrease postoperative Hb reduction and avoid allo-BT.

Several previous studies have shown that cardiovascular events are an important complication of TKA, and a leading cause of death [21,28,29]. Furthermore, an association between hemorrhagic anemia and cardiovascular events after total joint replacement has been reported [30]. No cardiovascular events occurred in our patients within 1 year after surgery. Lin et al. [20] reported that the incidence rate of ischemic heart disease was 1.13% (45 events among 3599 patients) within 1 year after primary TKA. Lu et al. [21] reported that the incidence rate of ischemic heart disease was 0.58% (80 events among 13,849 patients) within 1 year after primary TKA. Several previous studies have shown an association between knee OA and an increased risk of death [1,31]. Lovald et al. [1] reported that the mortality rate was 3.1% at 1 year after the diagnosis of knee OA in patients not undergoing surgery. Although the short-term mortality rate after TKA has been widely reported, few studies have focused on the long-term mortality rate after TKA [28,32,33]. Ohzawa et al. [32] reported that the survival rate was 88.7% within 5 years after TKA. In a study by Schroder et al. [33], the survival rate was 89% within 5 years after TKA and total hip arthroplasty. Lizaur-Utrilla et al. [29] reported that the survival rate was 89% within 4 years after TKA. In the present study, the mortality rate was 1.0% within 1 year after surgery, and 2.0% within 5 years after surgery. Furthermore, the incidence rate of ischemic heart disease within 1 year after surgery was 0%, although the risk among these patients was relatively high (mean age, 76.1 years; mean body mass index, 25.0 kg/m^2^). One reason for these low rates might be the avoidance of a rapid decrease in the Hb concentration during the perioperative period of TKA [34].

Among the 100 patients who underwent primary TKA with auto-BT in this study, only 1 required allo-BT, and no major complications associated with blood transfusion occurred. Dodd [3] reported that the infection rates associated with allo-BT were 1/225,000 for human immunodeficiency virus, 1/20,000 for hepatitis B virus, 1/150,000 for hepatitis C virus, and 1/50,000 for human T-lymphotropic virus. Goodnough et al. [35] reported respective incidence rates of febrile non-hemolytic transfusion reaction and hemolytic transfusion reaction of 3% and 0.0001%, respectively. Our findings suggest that our procedure was relatively safe in terms of complications.

To our knowledge, few studies have investigated cardiovascular events and the mortality rate in TKA using only auto-BT. Our findings suggest that auto-BT in primary TKA might be effective for safety and health outcomes. Although the rate of allo-BT in TKA is higher than that of auto-BT [18,19], blood donations and stock have seriously decreased because of the worldwide spread of COVID-19 [9,10]. To secure the blood stock for urgent blood transfusion, auto-BT may be effective in elective surgery such as TKA.

Several limitations of the present study warrant mention. First, this study was retrospective and used only 800 mL of autologous blood for transfusion. There is a lack of control series in the current study. Because we did not have control series in our hospitals, we compared the results in the current study with that of many previous reports to indicate the efficacy of our procedure for not only management of perioperative complications, but also mid-term health. Further study would be required to determine an association between Hb change and mid-term health. Second, the sample size was relatively small. However, we analyzed the Hb concentration more frequently than in previous studies (eight times in this study).

In the present study, primary TKA with preoperatively collected blood for auto-BT showed relatively small changes in the perioperative Hb concentration, with only 1% of patients requiring additional allo-BT. We also showed a low incidence of cardiovascular events and a low mortality rate within 1 and 5 years after surgery. These results suggest that auto-BT, in which blood is preoperatively collected, in patients undergoing TKA, might be useful for ensuring patient safety and health, even if its cost-effectiveness may be debatable.

## Figures and Tables

**Figure 1 jcm-11-02252-f001:**
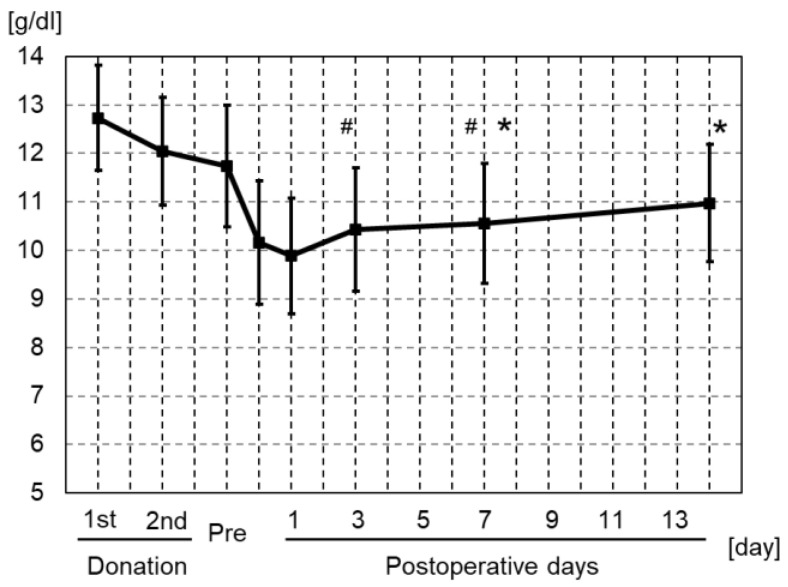
Mean Hb concentrations during the operative period for all patients. The lowest Hb concentration was 9.9 ± 1.2 g/dL on postoperative day 1. On postoperative day 14, the Hb concentration in that patient had recovered to 11.0 g/dL. The error bars indicate the standard deviation. * *p* < 0.05 compared with Hb of postoperative day 0. ^#^
*p* < 0.05 compared with Hb of postoperative day 1.

**Figure 2 jcm-11-02252-f002:**
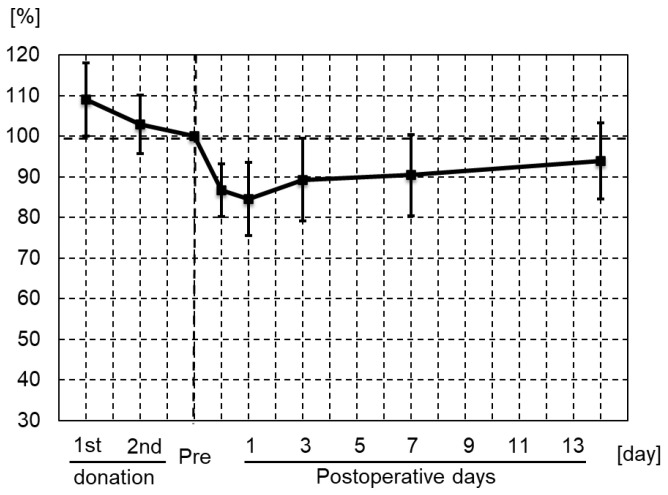
Percentage of the Hb concentration the day before the operation for all patients. The percentage of the preoperative Hb concentration was lowest on day 1 (84.5%), and recovered to 93.9% on day 14.

**Table 1 jcm-11-02252-t001:** Patient characteristics.

Age (Years)	76.4 ± 6.2 (55–81)
Sex (women/men)	89/11
Height (cm)	150.9 ± 7.2 (137.5–175)
Body weight (kg)	57.1 ± 10.1 (41.9–81.1)
BMI (kg/m²)	25.0 ± 3.6 (18.8–35.9)
Diagnosis (OA/RA)	94/6
Operating time (minutes)	101 (68–172)
Estimated blood loss during operation (mL)	526.0 ± 258.0
Estimated blood loss from post to 3 days (mL)	1235.8 ± 418.8

OA—Osteoarthritis; RA—Rheumatoid arthritis; data (mean ± standard deviation (range)).

**Table 2 jcm-11-02252-t002:** Hemoglobin concentration during the pre- and postoperative period.

		Hemoglobin Concentration (g/dL)
Donation	1st	12.7 ± 1.1
	2nd	12.0 ± 1.1
Preoperative		11.7 ± 1.2
Postoperative	Day 0	10.2 ± 1.2
	Day 1	9.9 ± 1.2
	Day 3	10.4 ± 1.3
	Day 7	10.5 ± 1.3
	Day 14	11.0 ± 1.2

**Table 3 jcm-11-02252-t003:** A comparison of the current study results with the literature.

Donation	Postoperative Lowest Hemoglobin			Blood Test Dates	Autologous Transfusion			Allo-BT Rate and Volume
	Concentration (g/dL)	DAS	RFP (g/dL)		Type	Rate	Volume	
Ha et al. [23]	8.9	2	4.0	Preoperative, postoperative day 1, 2, and last of follow-up	-	-	-	36.1% (13/36), 278.9 mL
	10.5	2	2.6		ATS	100% (35/35)	487 mL	11.1% (4/35) 64.4 mL
Cip et al. [14]	10.2	3	4.1	Preoperative, postoperative day 1, 3, and 5	-	-	-	33% (23/70), 2.1 units
	9.9	3	3.7		ATS	100% (70/70)	283 mL	33% (23/70), 2.1 units
Blazekovic et al. [4]	9.2	2	5.0	Preoperative, postoperative day 1, 2, and 5	ATS	NA	NA	33% (6/18)
	9.3	5	5.6		ATS and auto-BT (72 h ^a^)	NA and 61.1% (11/18)	NA	16.7% (3/18), NA
	9.3	5	5.2		ATS and auto-BT (2 wks ^b^)	NA and 55.5% (10/18)	NA	16.7% (3/18), NA
Cho et al. [24]	8.8	3	4.5	Preoperative, postoperative day 0, 1, 2, 3, 5, 7, and 14	-	-	-	Excluded patients receiving allo-BT
	9.2	3	3.7		IA TXA injection			
Chang et al. [25]	9.5	4	3.5	Preoperative, postoperative day 0, 1, 2, 4, and 6	IV TXA injection			2% (1/48)
	9.7	4	3.2		IV and 2 days of oral TXA			2% (1/46)
	9.1	4	3.4		IV and 5 days of oral TXA			2% (1/47)
Our study	9.9	1	1.8	Preoperative, postoperative day 0, 1, 3, 7, and 14	auto-BT	100% (100/100)	800 mL	1% (1/100)

DAS—Day after surgery; RFP—Reduction from preoperative; ATS—Autologous transfusion system (defined as BS in the current study); TXA—Tranexamic acid; IA—intra-articular; IV—intravenous. ^a,b^ The interval between donation and surgery.

## Data Availability

Data available upon request from D.M.

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
