# Peer review of "Effects of Pre-Donated Autologous Blood Transfusion on Peri-Operative Hemoglobin Concentration and Mid-Term Health Outcomes in Primary Total Knee Arthroplasty"

_jcm, 2022, doi:10.3390/jcm11082252_

Round 1

Reviewer 1 Report

Thank you for the opportunity to review this manuscript. This is an interesting topic. Auto BT has been a well -known blood conservation technique in many other surgeries that are prone to increased surgical bleeding. Utilizing the concept of auto BT to TKA may be unique and worth considering. Please see my suggestions below.

 Summary:  Authors conducted a retrospective study reviewing perioperative management of 100 patients who received TKA. Auto BT was collected preoperatively and erythropoiesis was stimulated by treating them with erythropoietin in the preoperative period in between auto BT collection. Study demonstrated that auto BT has slowed down reduction postop hemoglobin and also the lowest level of Hgb was least compared to other studies. Only 1% (1 patient) required allo BT.

 General assessment of the paper

Overall, it was a well-designed paper. Auto BT definitely has shown to improve postop outcomes. Authors have conducted a thorough search and discussed previous studies in the discussion to demonstrate the strength of their study. Subjects’ numbers could have been higher but authors have already mentioned that as one of the limitations. Please see my comments below to consider revising the manuscript.

 Specific comments/critiques

 Introduction –

L 53 - , “ BS can reportedly increase the postoperative blood drainage [15,16].” _ please elaborate how – dilutional coagulopathy? BS consists of mostly RBCs after filtration – all the clotting factors and platelets are washed away.

L 55 - rate of blood-borne infection – Chances of infection/contamination of auto BT is not minimal – this statement may give a false impression that infection is not a risk with auto BT. I’d carefully rephrase the sentence.

Material & Method –

Auto BT collection – since it is a retrospective study, I’d suggest to rewrite this paragraph in passive voice. For example –

L 90 - Instead of writing, ‘we collected 400 cc …’, consider writing ‘Autologous blood (400 cc) was collected from the patients’.

This will give a more realistic sense of retrospective study which is basically reviewing the charts of the patients who had received some interventions as per standard protocol and authors are collecting data to see impact of the interventions. The way it has been described now sounds more like a prospective study where auto BT was implemented to see the effect.

Perioperative blood tests –

Same as above. Please consider rewriting as data was collected from blood tests that were done routinely on every postoperative patient who met eligibility criteria for the study.

Results –

Please explain Table 1 more explicitly. Add an explanatory comment below the table describing right column of the table [e.g mean +/- SD; (range)]

Discussion –

L 217 – While describing advantages of auto BT over BS, please explain that BS only provides RBC supplement, and all the clotting factors and platelets are washed away. This in turn sometimes may attribute to dilutional coagulopthay from large volume BS transfusion.

L235 – This economic aspect of auto BT was very important. It is also worth reinforcing, risks of allogenic BT and management of complications arising from allogenic BT may be expensive as well.

Reviewer 2 Report

  1. Error of "fin" in line 134, and it should be "in" 
  2. The description of "mortality rate of 1.0% and 2.0%, respectively" on line 160-160 is not clear enough, and this important result should be in a separate line to clearly point out your excellent performance.
  3. Since all patients received edoxaban for 2 weeks from the day after the operation, it would be more comprehensive if a DISCUSSION section focus on the effect of edoxaban in your study. 

Author Response

This manuscript is a resubmission of an earlier submission. The following is a list of the peer review reports and author responses from that submission.

Round 1

Reviewer 1 Report

Summary

The authors are presenting a retrospective study on the outcome of patients receiving autologous blood transfusions in primary TKA. In addition, the authors present the cardiovascular events and the mortality rate in the first 5 years postoperatively.

The authors discuss an important topic of total knee arthroplasty.

Comments

Title

The title is accurate.

Abstract

The abstract is accurate.

Material and Methods

This section has major flaws. First the author do not state how the postoperative data of the patients were collected. Did all patients return for the follow up to the hospital or did they contact them via email or telephone? The second missing point is the statement how many patients were lost to follow up. This is a important statement for the reader in order to classify the results.

How where the patients chosen to be part of this study? The inclusion and exclusion criteria are missing.

Results

The results are accurate.

Discussion

The discussion section is appropriate.

Tables

The tables are accurate.

Figures

The figures are appropriate.

Summary

In summary I think that the theme of the article is very interesting. But I think that there are several flaws concerning the material and methods so the paper should undergo a major revision.

Reviewer 2 Report

I don't see this study suitable for different reasons:

  • sample size too small for purpose of the study
  • too many variables not considered that can influence HB levels
  • blood collection too close to surgery
  • no control group
  • complications at 1 and 5 years cannot be related to transfusions 
  • no statistical analysis